# Novel Insights into the Nature of Intraspecific Genome Size Diversity in *Cannabis sativa* L.

**DOI:** 10.3390/plants11202736

**Published:** 2022-10-16

**Authors:** Manica Balant, Roi Rodríguez González, Sònia Garcia, Teresa Garnatje, Jaume Pellicer, Joan Vallès, Daniel Vitales, Oriane Hidalgo

**Affiliations:** 1Institut Botànic de Barcelona (IBB, CSIC-Ajuntament de Barcelona), Passeig del Migdia s.n., 08038 Barcelona, Catalonia, Spain; 2Royal Botanic Gardens, Kew, Kew Green, Richmond TW9 3AE, UK; 3Laboratori de Botànica (UB), Unitat Associada al CSIC, Facultat de Farmàcia i Ciències de l’Alimentació–Institut de Recerca de la Biodiversitat (IRBio), Universitat de Barcelona, Av. Joan XXIII 27–31, 08028 Barcelona, Catalonia, Spain

**Keywords:** Cannabaceae, *Cannabis sativa*, genome size, intraspecific genome size variation, population variability, sex chromosomes

## Abstract

*Cannabis sativa* has been used for millennia in traditional medicine for ritual purposes and for the production of food and fibres, thus, providing important and versatile services to humans. The species, which currently has a worldwide distribution, strikes out for displaying a huge morphological and chemical diversity. Differences in *Cannabis* genome size have also been found, suggesting it could be a useful character to differentiate between accessions. We used flow cytometry to investigate the extent of genome size diversity across 483 individuals belonging to 84 accessions, with a wide range of wild/feral, landrace, and cultivated accessions. We also carried out sex determination using the MADC2 marker and investigated the potential of flow cytometry as a method for early sex determination. All individuals were diploid, with genome sizes ranging from 1.810 up to 2.152 pg/2C (1.189-fold variation), apart from a triploid, with 2.884 pg/2C. Our results suggest that the geographical expansion of *Cannabis* and its domestication had little impact on its overall genome size. We found significant differences between the genome size of male and female individuals. Unfortunately, differences were, however, too small to be discriminated using flow cytometry through the direct processing of combined male and female individuals.

## 1. Introduction

*Cannabis sativa* L. (hereafter referred to as *Cannabis*) is one of the most versatile plants used by humans over millennia. Despite being mostly known for its psychoactive use, *Cannabis* has played an important role in everyday life for hundreds of years. For example, it was extensively used in traditional medicine and became an important source of fibre and food [1]. However, as a consequence of its illegal status, the use of *Cannabis* was abandoned in many parts of the world. Nonetheless, in recent years, the cannabis industry has experienced a rising interest beyond its recreational uses, including more sustainable options in textile, automotive, construction, food, and cosmetic applications [1,2,3,4]. 

The genus most likely originated in the NE Tibetan Plateau more than 25 Mya [5,6], from where it is thought to have spread to North and West Asia and Europe, before continuing to expand eastwards and southwards [5]. Genetic and archaeological evidence suggests that the domestication of *Cannabis* took place approximately 12,000 years ago in East Asia. It was used as a multipurpose crop until c. 4000 years ago, when separate selections for fibre and drug production started [7]. Since then, large-scale cultivation as a crop has enabled its spread around the world, and today, *Cannabis* has a worldwide distribution [8,9]. 

The wealth of different applications through centuries resulted in the development of a wide range of cultivars, varieties, and strains adapted to different climates with high morphological and phytochemical diversity [10]. Depending on the cultivation purpose, morphology, and chemical composition, domesticated *Cannabis* can be separated into fibre-type (namely hemp; <0.3% Δ9-tetrahydrocannabinol (THC)) or drug-type (marijuana and medicinal *Cannabis*; >0.3% THC) plants [11]. Within the drug-type plants, different chemotypes are recognised based on their chemical profiles, which are mainly underpinned by the differences in THC/CBD (cannabidiol) ratios. Recently, other secondary metabolites (such as terpenoids and flavonoids) have also gained an important role [12,13,14]. The morphological and chemical diversity of *Cannabis* has hampered its taxonomic resolution, leading to different taxonomic treatments over the years (see McPartland and Small [15] for a detailed review). Currently, it is considered a monotypic genus, with *C. sativa* as the only accepted species. However, according to a recent evolutionary study based on whole-genome resequencing, fibre-type and drug-type cultivars constitute distinct genetic lineages that diverged from an ancestral gene pool, currently represented by wild or naturalised plants in Central and East Asia, which could have taxonomic implications [7].

Genome size (or C-value) is defined as the amount of DNA in the holoploid genome of an individual [16], and is considered to be relatively constant within a species [17]. Despite reports of intraspecific genome size variation having long been treated with caution, the advent of high-resolution techniques for genome size estimation, such as flow cytometry, has provided strong evidence of intraspecific variability in several taxonomic groups. In general, such variation has been attributed to, e.g., hybridisation and/or polyploidisation events [18,19], B-chromosomes [20], heteromorphic sex chromosomes [21,22], changes in non-coding repetitive DNA [23], presence/absence of specific DNA sequences [24], and illegitimate recombination [25]. In addition to that, intraspecific genome size variation has also been related to extrinsic and/or abiotic factors such as altitude [26,27,28,29,30], latitude [24,31,32,33], and temperature [31], and to different phenological and morphological characters [27,34].

*Cannabis* is an annual, wind-pollinated, dioecious plant, although some monoecious cultivars also exist [35]. The diploid genome generally presents 20 chromosomes, 18 autosomal chromosomes and one pair of sex chromosomes. Female and monoecious plants have two X chromosomes (XX), while male plants have heteromorphic X and Y (XY) chromosomes [36,37]. Multiple studies investigating genetic [7,38,39,40], morphological [41,42,43], and phytochemical diversity [12,44,45,46,47] in *Cannabis* have been published, however, only five of them included genome size measurements [37,48,49,50,51]. Most of these studies were carried out on cultivars and centred on either detecting polyploids, or differences in genome size between individuals of different sexes. Certainly, only the study by Lee et al. [50] focused on intraspecific genome size variation in *Cannabis*. These authors detected differences between accessions of different origins, suggesting that genome size could be used as a character to discriminate among accessions. Despite this, intraspecific variability in the genomic content of *Cannabis* has continued to receive little attention. With regard to ploidy levels, natural polyploidisation in *Cannabis* has only been reported once so far, in a wild tetraploid population from India [52]. Small [53] analysed over 200 accessions and found all of them to consistently be diploids (2n = 20). However, artificial polyploids can be induced under laboratory conditions (e.g., chemical treatments), and indeed, triploid, tetraploid, and mixoploid *Cannabis* plants have been produced in plant breeding programs [11,51,54,55,56,57]. 

Many efforts have been made to develop tools to discriminate between male and female *Cannabis* individuals, some of them involving genome size. Although the exact mechanism underpinning sex determination in the species is not yet fully comprehended [8,58], it is thought to be determined by an XY chromosome pair [36,49,59] or by the X to autosome ratio [37,60]. Since the Y chromosome is slightly longer than the X chromosome, male individuals are expected to present a larger genome size. This was corroborated by studies that have found a difference between sexes of ∆ = c. 0.05 pg/2C [37,49] or even up to ∆ = 0.15 pg/2C [50]. Early sex determination is usually carried out using male-associated DNA markers [61,62,63,64,65,66,67], but the accuracy and reproducibility of some of them have been questioned [67,68]. Based on the above, there is no doubt that developing a method of sex detection through flow cytometry, as previously suggested [50], would be of great interest. However, the reliability and limitations of the method are still to be evaluated for *Cannabis*.

The worldwide distribution of *Cannabis*, its large morphological and phytochemical variability, the existence of heteromorphic sex chromosomes, and the fact that the plant has been a target for selection by humans, could be reflected (to some extent) at the genome size level. So far, most of the studies have focused on a few different (either fibre or drug) *Cannabis* cultivars, but very rarely wild accessions were included. Here, we gathered a large number of wild/feral, landrace and cultivar *Cannabis* accessions, covering a wide distribution area in order to (i) evaluate the extent of genome size and ploidy level diversity in the species; (ii) investigate how this diversity distributes across accessions, geographical ranges, and sexes; and (iii) test whether flow cytometry can be used as a standard tool to distinguish between male and female *Cannabis* individuals in both wild/feral and cultivated accessions.

## 2. Results and Discussion

### 2.1. Genome Size in *Cannabis*: Evidence of Intraspecific Variation

We analysed 483 individuals belonging to 84 accessions (i.e., populations of wild/feral plants, or any landrace and cultivar) from an area spreading over more than 12,000 km and three continents (Figure 1, Appendix A). Nuclear DNA content (2C-values) obtained per individual and summarised by accession and geographical region are depicted in Figure 1 and Appendix A.

All but one of the individuals analysed were diploid, with genome sizes ranging from 1.810 pg/2C (individual Mongolia 5.14) up to 2.152 pg/2C (individual Armenia 15.3), and an average of 1.956 ± 0.051 pg/2C. One triploid individual was found in a North-Indian wild accession, with a genome size of 2.884 pg/2C. Illustrative flow cytometry histograms for diploid and triploid individuals are presented in Figure 2A,B. The average genome size value for diploid *Cannabis* accessions obtained in our study is slightly higher than average values previously reported (1.720 pg/2C, range = 1.42–1.97 pg/2C; Appendix A; [37,48,49,50,51]). These differences could be explained by the use of different internal standards, instruments, and stains [69]. 

The overall genome size difference between diploid individuals spanned over a 1.189-fold range (18.89%). We illustrated for the first time the intraspecific variation in *Cannabis* by processing samples with different genome sizes together and obtaining two peaks (Figure 2C). It is to note that the variation we highlighted through the analyses of 482 diploid individuals is much smaller than the one previously obtained by Lee et al. [50]. Indeed, these authors found a 1.373-fold (37.3%) intraspecific difference through the analysis of 35 individuals.

At the accession’s level, we detected significant differences in genome size of diploids across the 84 analysed accessions (*p* < 0.001, Table 1), with average 2C-values ranging from 1.890 ± 0.053 pg/2C (Romania 8) to 2.028 ± 0.022 pg/2C (Armenia 1), which represented a 1.073-fold variation (7.3%). Lee et al. [50] found, however, a much larger variability (1.36-fold range; 35.9%), although they analysed only 14 accessions, with 2C-values ranging from 1.42 to 1.93 pg/2C. In turn, Faux et al. [37] did not find a significant difference among the genome sizes of five *Cannabis* monoecious cultivars. The variation within accessions in our dataset ranged from 1.020-fold (∆ = 0.038 pg/2C, Romania 4) up to 1.123-fold (∆ = 0.236 pg/2C, in Armenia 15), with an average of 1.053-fold (∆ = 0.101 ± 0.032 pg/2C) (Figure 1C, Appendix A). Similarly, the study by Lee et al. [50] detected a within-accession variation from 1.006-fold (∆ = 0.01 pg/2C) to 1.127-fold (∆ = 0.22 pg/2C). We found a significant difference in genome size across accessions and distribution areas (Figure 1; ANOVA, *p* < 0.001, Table 1), however, no accession nor area could be clearly separated from the rest through the Tukey HSD post hoc test. 

Taking together these results, despite the differences in the degree of genome size variation when compared with previous studies, our results provide compelling evidence of genuine intraspecific variation in *Cannabis*.

### 2.2. Potential Factors Influencing Genome Size Variation in *Cannabis*

Intraspecific genome size variation of taxa with a large distribution area or isolated populations has been mostly attributed to changes in ploidy level, though, cases of intraspecific variation at the same ploidy level as found in *Cannabis* have also been reported, such as in *Urtica dioica* (2*x* and 4*x* populations with 3.05% within 2*x* accessions and 9.8% variability within 4*x* accessions [33]), *Festuca pallens* (2*x* and 4*x* populations with 16.6% variation in 2*x* and 15% in 4*x* [70]), *Picris hieracioides* (37.6% variability [71]), *Senecio carniolicus* (13.1% variability in 2*x*, 10.2% in 4*x*, 5.4% in 5*x*, and 10.5% in 6*x* populations [72]), *Ranunculus parnassifolius* (2*x* populations with 8.58% and 4*x* with 1.29% variability [73]), and *Euphrasia arctica* (27.4% variability in 2*x* accessions [24]). Intraspecific genome size variation in species with characteristics comparable to *Cannabis*, i.e., a large distribution area and/or the presence of numerous cultivars, has also been reported in *Chenopodium album* (Europe–China; 6.13% [74]), *Chenopodium quinoa* (Americas; 5.9% [75]), *Prunus armeniaca* (Europe–China; 2.3% [76]), and *Cardamine occulta* (Europe–Japan; 8.98% [77]). The intraspecific variation in genome size we found in diploid *Cannabis* at the level of the individuals (18.89%) and accessions (7.3%) is, therefore, similar to that found in other taxa. 

Given that no differences in chromosome numbers—except for a few cases—have been found in *Cannabis* (see [50,53] and Appendix A), the variation we observed is unlikely to be caused by aneuploidy (i.e., changes in chromosome number). *Cannabis* has heteromorphic sex chromosomes [49], therefore, the sex of individuals could account for some of the variation in genome size. Even though most of our analysed dioecious accessions included both male and female individuals, their frequencies within accessions were not always the same, which could affect the average genome size values per accession. However, according to our results, sex does not fully explain the variation detected between accessions (further discussed below). In the absence of chromosome number variation, another possible explanation for intraspecific genome size variation could be the differences in repetitive DNA sequence content. Pisupati et al. [78] found that 64% of the *Cannabis* genome is made up of repetitive sequences. This is less than in *Zea mays* (c. 85% [79]), but more than in *Arabidopsis thaliana* (c. 21% [80]), where intraspecific genome size variation has also been found [29,81]. Finally, although we have made a great effort to optimise the method for genome size assessment in *Cannabis* by testing a wide range of plant tissues, growing stages, and nuclei extraction buffers (see Section 3. Materials and Methods), we cannot entirely rule out that part of the variation could be due to a technical error. Indeed, all *Cannabis* parts are very rich in secondary metabolites [44], and previous studies have shown that chemical compounds can interfere with DNA binding of the stain, thus, potentially altering the genome size assessments [82,83,84,85,86,87]. However, we are confident that we have minimised this effect by using only very young leaves from newly germinated seedlings, which provided the best quality measurements in our preliminary tests.

### 2.3. Events of Polyploidy in *Cannabis* Are Extremely Rare

We found one triploid and 482 diploid individuals (Figure 1, Appendix A). These results are similar to the previous evidence of Small [53] and Lee et al. [50], showing consistent diploidy (with minimal exceptions) in the species. We confirmed chromosomally that the ploidy levels inferred with flow cytometry by carrying out chromosome counts in 10 individuals from 10 accessions. We found 2n = 20 in diploids and 2n = 30 in the triploid individual (wild North-Indian accession IND1; Figure 3; Appendix A). This is the first report of a wild-born triploid individual in *Cannabis*. Records of non-diploid *Cannabis* individuals were indeed so far limited to a tetraploid population in North India [52], or they were otherwise induced by chemical treatment [51]. From the same accession as the triploid individual, the genome size of three other individuals was measured—they were all diploids. The triploid was a male, had a similar morphology than other individuals, and it flowered normally. Unfortunately, we were not able to study this accession further due to the limited number of seeds available, but it would certainly be interesting to investigate whether other ploidy levels could be found in this or more accessions. 

Our results confirm that natural polyploidy seems to be extremely rare or even practically non-existent in *Cannabis*, despite its rich domestication background. This contrasts with evidence found in many other species, where genome polyploidisation is preceding or concomitant with their domestication [88,89]. Whole genome multiplication and subsequent diploidization processes provide plants with increased allelic diversity, heterozygosity, and enhanced meiotic recombination, which may increase their adaptive plasticity and evolutionary success [89]. It is, therefore, not surprising that the domestication of some of the most economically important cultivated plants is associated with a polyploidization event, e.g., *Avena sativa* [90], *Triticum* sp. [91], *Ipomoea batatas* [92], *Brassica rapa* [93], and *Musa* sp. [94], among others. In *Cannabis*, artificial polyploids have been obtained by several breeding programs; however, the changes in morphology and phytochemistry of the polyploids have not been extensively investigated so far, thus, requiring more research to be carried out [95].

### 2.4. Differences in Genome Size Values of Male and Female *Cannabis* Individuals

From the 99 individuals with the previously measured genome size selected for sex determination, a MADC2 male-associated band of 390 bp amplified in 46 of them (considered males), while the male-associated band was absent in 49 (considered females). Four individuals showed inconclusive results, with either no PCR bands or two non-indicative bands. 

The average female genome size was 1.947 ± 0.065 pg/2C (1.810–2.152 pg/2C), and the average male genome size was 1.987 ± 0.0521 pg/2C (1.920–2.112 pg/2C) (Figure 4 and Appendix A; Table 2). Using ANOVA, we found a significant difference in genome size between male and female plants (*p* < 0.001) (Figure 4, Table 1). The 2C-value of male individuals was in general larger than females for ∆ = c. 0.050 pg (0.0009–0.114 pg), which agrees with previous studies [37,49,50]. However, we found few cases where within the same accession, male individuals had a smaller genome size than females. Additionally, the overlap of genome size values of male and female individuals within accessions was, in general, quite high (Appendix A). 

*Cannabis* is showing significant genome size differences between male and female individuals, which is not always the case in dioecious species (e.g., in *Juniperus thurifera* [96]). The presence of a larger genome size in males has been reported in most plant species with heteromorphic sex chromosomes. While some dioecious species have differences in genome size between male and female individuals of similar magnitude to those found in *Cannabis* (2.05%), e.g., 0.45% in *Simmondsia chinensis* [97], 1.97% in *Viscum album* [97], and 2.09–4.19% in *Silene latifolia* [97,98], other species present much larger differences, e.g., 7.14% in *Rumex acetosa* [99], 9.83% in *R. hastatulus* [100], and 10% in *Coccinia grandis* [101]. A larger genome size in male is probably related with Y chromosome degeneration in plants, likely involving the accumulation of repeats in this non-recombining chromosome, as found in *R. acetosa* [102], *Cannabis*, and some *Humulus* species [103]. 

### 2.5. Sex Determination in *Cannabis* Using Flow Cytometry

Peaks of male and female *Cannabis* individuals from the same accession analysed together through flow cytometry overlapped in all cases. This can be explained by the fact that the largest difference between male and female individuals we intended to discriminate was ∆ = 0.076 pg/2C (Armenia 3; Appendix A), which is well below the smallest genome size difference for which we obtained distinguishable fluorescence double peaks in *Cannabis* (i.e., ∆ = 0.130 pg/2C). Our results showed that while differences between the genome size of male and female individuals are significant (according to ANOVA; see part 2.3 for more details), they are simply too small to be discriminated using flow cytometry, by directly processing together male and female individuals. In previous reports, the differences between male individuals on the one hand, and female and monoecious individuals (in both the sex is determined by two X chromosomes) on the other, detected by Faux et al. [37] and Sakamoto et al. [49] (∆ = 0.046 pg/2C and ∆ = 0.048 pg/2C, respectively), were also extremely small. Only Lee et al. [50] found larger differences of ∆ = 0.05–0.15 pg/2C (2.90–10.56%) between sexes, that could potentially be discriminated in flow cytometry histograms. Unfortunately, the individuals demonstrating these large differences were not processed together to confirm these results. It should be noted, however, that our results were obtained using propidium iodide as the dye in the flow cytometry experiments. Certainly, other methods of flow cytometry, such as the use of other fluorochromes (for example DAPI) or flow sorting, that could offer an improved resolution limit of the technique, should be explored in the future for inexpensive and high-throughput early sex determination in *Cannabis*. Indeed, a previous study has shown the suitability of DAPI flow cytometry for direct sex identification in *Silene latifolia* (formerly *Melandrium album*) and *Silene dioica* (formerly *M. rubrum*) [22], allowing for the discrimination of approximately 1.04-fold genome size difference.

## 3. Materials and Methods

### 3.1. Plant Sampling and Cultivation

We analysed 483 *Cannabis* individuals from 84 accessions distributed worldwide, spanning over 12,000 km (Appendix A). On average, 5 individuals from each accession were analysed (see Appendix A for details on specific accessions). Seeds from the studied accessions were germinated in Petri dishes and transplanted to pots after the emergence of the first leaves. Plants were cultivated in a growth chamber under controlled conditions (25 °C, 18 h light/6 h dark). Studied individuals were grown for approximately 2–3 weeks until the development of the first or second pair of leaves. 

### 3.2. Flow Cytometry Measurements

Genome size was determined using a CyFlow Space instrument (Sysmex-Partec GmbH, Goerlitz, Germany), fitted with a 100 mW green solid-state laser (Cobolt Samba, Cobolt AB, Solna, Sweden). The internal standard *Petroselinum crispum* ‘Champion Moss Curled’ (2C = 4.50 pg) [104] was used. 

*Cannabis* plants have many secondary metabolites [44] that could potentially interfere with DNA staining and worsen the quality of the measurements. To overcome such potential issues, different plant tissues and growing stages were tested. The best results were obtained using the first or second pair of leaves of young *Cannabis* plants. Additionally, different flow cytometry buffers (LB01 [105], Ebihara [106], Cystain Ox Protect and PI Absolute buffers (Sysmex-Partec GmbH)) were tested as well, before choosing the general purpose buffer GPB [107] supplemented with 3% PVP-40 [108] as the most appropriate one. Additional measures, such as reducing chopping intensity and working in ice-cold conditions, were taken to reduce the potential effects of secondary metabolites.

We followed the one-step procedure [109] with some modifications. Fresh leaf samples of *Cannabis* and the standard were co-chopped in a Petri dish over ice using 2 mL of the selected nuclei extraction buffer. The sample was then filtered, stained with 40 µL of propidium iodide (PI), and vortexed; samples were left on ice for approximately 30 min before the measurement. 

For each sample, the nuclear DNA content was estimated by counting approximately 1000 nuclei per fluorescence peak. Each sample was assessed two times and the results averaged to obtain the final genome size value for the individual. The histograms were analysed using the FlowMax software (v. 2.9, Sysmex-Partec GmbH). Histograms with coefficients of variation (CVs) larger than 5% were discarded. 

### 3.3. Chromosome Counts

Root meristems from each accession were collected for chromosome counts, pre-treated for 2.5 h in 0.05% aqueous colchicine and fixed in fresh absolute ethanol and glacial acetic acid (3:1) for 3 h at room temperature, before being stored in the fixative at 4 °C. They were hydrolysed for 10 min at 60 °C in 1N HCl and stained in 1% aqueous aceto-orcein for at least two hours. Root tips were subsequently squashed in a drop of 45% acetic acid-glycerol (9:1) and observed with a Zeiss Axioplan microscope (Carl Zeiss, Oberkochen, Germany). Metaphases were photographed using a Zeiss AxioCam HRm camera (Carl Zeiss). 

### 3.4. Sex Determination Using Male-Associated Marker and Flow Cytometry

To address the potential differences in genome size between male and female individuals, leaf material from 15 accessions (99 individuals) (Appendix A) was collected after genome size measurements and stored in silica gel. DNA was extracted either using the E.Z.N.A. SP Plant DNA Kit (Omega Bio-Tek, Norcross, GA, USA) or the CTAB protocol, following the method by Doyle and Doyle [110] with some modifications. 

The sex of individuals was tested using a male-associated DNA marker MADC2, with sequences 5′-GTGACGTAGGTAGAGTTGAA-3′, corresponding to the positions 1–20, and 5′-GTGACGTAGGCTATGAGAG-3′, corresponding to the positions 373–391 [62]. PCR reactions were performed in a 25 µL reaction mixture, containing 1 µL of genomic DNA (approximately 50 ng), 14.3 µL of sterile water, 2.5 µL of 2 mM MgCl2, 2.5 µL of 10X Gene Taq Universal buffer (Applied Biosystems, Carlsbad, CA, USA), 2.5 µL of 2.5 mM dNTPs mixture, 1 µL of each primer (5 pmol/µL), and 0.2 µL of AmpliTaq DNA polymerase (Applied Biosystems, Carlsbad, CA, USA). The amplification was carried out following the steps: 94 °C for 5 min followed by 37 cycles of 94 °C for 30 s, 58 °C for 1 min, 72 °C for 1 min, and a final step of 72 °C for 5 min. PCR products and ladder (HyperLadder™ 100 bp; Meridian Bioscience, Cincinnati, OH, USA) were separated on 2% agarose gels stained with SYBR Safe-DNA Gel Stain (Thermo Fisher Scientific, MA, USA), and were run at 100 V. 

As the reliability of the MADC2 marker used here has been questioned in the past, we first tested the marker on 43 individuals of wild/feral, landrace, and cultivar *Cannabis* accessions with previously known sex (plants grown until the reproductive phase). The marker proved to be a reliable method to assign the correct sex in all but one case, which was inconclusive. No false positives were detected. 

To test the suitability of flow cytometry to discriminate between male and female *Cannabis* plants, we selected five accessions (Appendix A) displaying a particularly wide range of genome sizes in a preliminary genome size survey (Appendix A). New plants from these accessions were cultivated. The first leaf of all individuals was collected and dried in silica gel, and this material was then used to detect the sex-associated marker MADC2 as described above. The genome size was determined by flow cytometry. Samples of each sex from the same or different accessions showing the most divergent genome size values were processed together to test whether genome size differences were large enough to be detected directly by flow cytometry (presence of double peaks).

### 3.5. Statistical Analyses

To analyse genome sizes across different accessions and distribution areas, we used the dataset composed of all 482 diploid individuals from 84 accessions (Appendix A). We analysed the differences using the analysis of variance (ANOVA). The difference in genome size between male and female individuals was also analysed using ANOVA on a dataset of 96 individuals from 15 accessions for which the sex was previously determined with the MADC2 marker (95 individuals); one additional individual where the MADC2 marker showed inconclusive results, but rapidly reached the reproductive phase, was also included (Appendix A). Before performing the ANOVA tests, the normality of the datasets was tested on residuals using the Shapiro–Wilk test and Q-Q plots, and homogeneity of variances with Bartlett’s test. All the analyses and data visualisations were performed using R version 4.2.1 [111].

## 4. Conclusions

This study evidenced the extent of intraspecific genome size variation in *Cannabis* and its distribution between and within accessions in an extended sampling covering a wide range of wild/feral, landrace, and cultivated accessions. Our results suggest that the geographical expansion of *Cannabis* and its domestication had little impact on its genome size. In this sense, the pattern observed for genome size is similar to that of other traits in *Cannabis* (e.g., leaf and inflorescence phenotype): a high variability of difficult interpretation, as it does not seem tightly related to its geographical distribution or to infraspecific taxonomic differentiation. Consequently, further studies will be needed to confidently determine whether the observed pattern is a consequence of the history of *Cannabis*, tightly linked to humans, or an intrinsic characteristic of the species.

## Figures and Tables

**Figure 1 plants-11-02736-f001:**
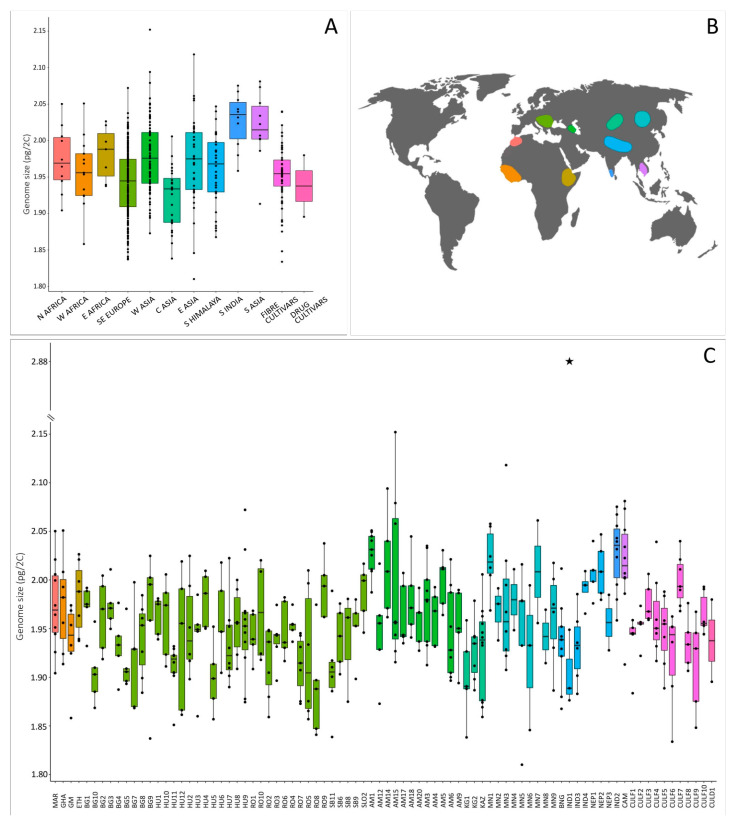
(**A**) Boxplots showing the distribution of genome size in diploid *Cannabis* individuals in different distribution areas. (**B**) Map of the areas of origin of the sampled accessions. (**C**) Boxplots showing the distribution of genome size in *Cannabis* individuals per accessions (the star indicates the genome size of the triploid individual found in the accession IND1—North India).

**Figure 2 plants-11-02736-f002:**
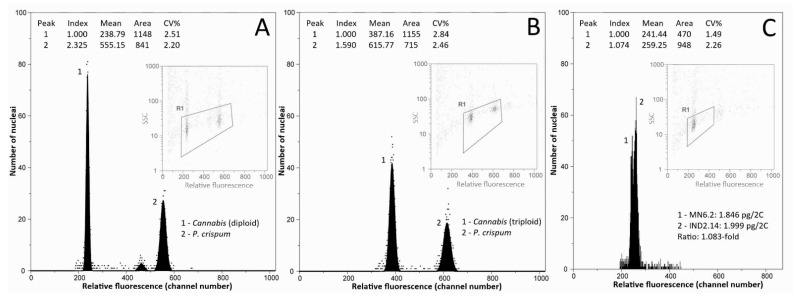
Flow histograms obtained from analysing (**A**) diploid *Cannabis* individual (accession KAZ, Kazakhstan) (peak 1) and (**B**) triploid *Cannabis* individual (accession IND1, North India), using *Petroselinum crispum* (4.5 pg/2C, peak 2) as the internal standard. (**C**) Flow histogram obtained from co-processing diploid individuals from accessions MN6 (Mongolia) and IND2 (South India).

**Figure 3 plants-11-02736-f003:**
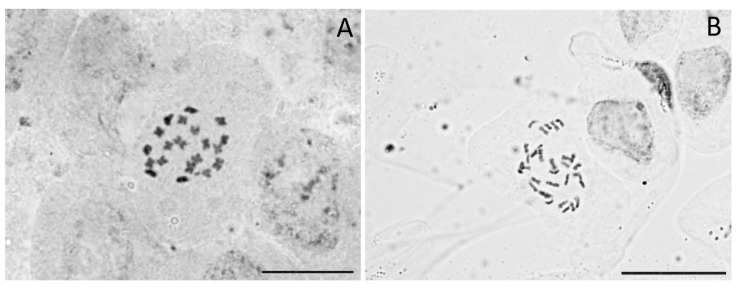
Somatic metaphase plates of a diploid *Cannabis* individual from the accession IND4—North India (2n = 20) (**A**) and a triploid individual from the accession IND1—North India (2n = 30) (**B**). Scale bars = 10 µm.

**Figure 4 plants-11-02736-f004:**
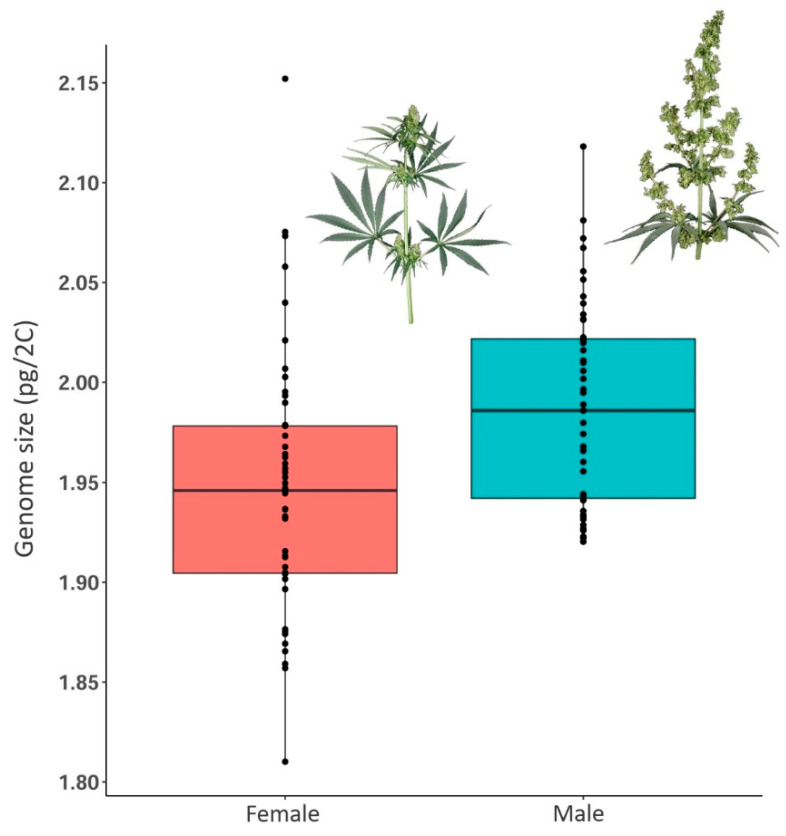
Boxplots showing the genome size distribution of female and male *Cannabis* individuals.

**Table 1 plants-11-02736-t001:** Results of ANOVA analysis comparing the effect of accessions, distribution areas, and sex on genome size values of *Cannabis*.

Variable	No. ind.	DF	Sum Sq.	Mean Sq.	F Value	*p* Value
**Accessions**	482					
Accession		83	0.5206	0.006272	3.386	<0.001
Residuals		398	0.7372	0.001852		
**Distribution area**	482					
Distribution area		11	0.2185	0.019863	8.983	<0.001
Residuals		470	1.0393	0.002211		
**Sex**	96					
Sex		1	0.0397	0.03965	11.62	<0.001
Residuals		94	0.3208	0.00341		

**Table 2 plants-11-02736-t002:** Differences in genome size between male and female individuals in the 15 selected *Cannabis* accessions. More details of the accessions can be found in the Appendix A.

Female Genome Size (pg/2C)						Male Genome Size (pg/2C)					
Accession	No. ind.	Mean	SD ^1^	Min.	Max.	No. ind.	Mean	SD ^1^	Min.	Max.	Difference
**AM15**	4	2.021	0.106	1.916	2.152	3	1.942	0.014	1.927	1.956	0.079
**AM3**	3	1.927	0.013	1.913	1.937	1	1.989	/	1.989	1.989	0.061
**BG3**	1	1.950	/	1.950	1.950	2	1.986	0.036	1.960	2.011	0.036
**CAM**	4	1.999	0.066	1.913	2.073	6	2.029	0.034	1.986	2.081	0.030
**HU11**	3	1.913	0.017	1.902	1.932	2	1.923	0.000	1.922	1.923	0.010
**HU9**	6	1.939	0.034	1.875	1.968	4	1.997	0.068	1.920	2.072	0.058
**IND2**	2	2.035	0.057	1.995	2.075	6	2.036	0.030	1.980	2.067	0.001
**CUL7**	5	2.003	0.027	1.973	2.040	2	1.989	0.030	1.968	2.011	0.014
**KAZ**	6	1.914	0.045	1.869	1.963	5	1.957	0.031	1.933	2.006	0.043
**MAR**	3	1.938	0.031	1.904	1.964	2	1.973	0.067	1.926	2.021	0.036
**MN3**	2	1.927	0.028	1.908	1.947	5	1.997	0.080	1.922	2.118	0.069
**MN5**	4	1.925	0.080	1.810	1.979	1	2.016	/	2.016	2.016	0.091
**RO2**	2	1.882	0.032	1.859	1.905	0	/	/	/	/	/
**RO3**	1	1.897	/	1.897	1.897	5	1.947	0.016	1.932	1.974	0.051
**RO5**	4	2.021	0.106	1.916	2.152	3	1.942	0.014	1.927	1.956	0.079

^1^ SD: standard deviation.

## Data Availability

The data presented in this study are available in the Appendix A.

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
