# Peer review of "Novel Insights into the Nature of Intraspecific Genome Size Diversity in *Cannabis sativa* L."

_plants, 2022, doi:10.3390/plants11202736_

Round 1
Reviewer 1 Report
Balant et al investigate the Genome Size 2 Diversity in Cannabis sativa L. over a very large distribution area, including wild and cultivated respectively introduced accessions. They found a high consistency in genome size. Despite significant differences in genome size between males and females, they could not detect it with the applied flow cytometric methods.
The benefits of the study are the large number of individuals investigated and – in case of the male-female discrimination -the clear evidence that the standard flow cytometry method is not applicable. The wording and structure of the article is clear and precise.
In my opinion, this is a useful study.
Reviewer 2 Report
It will be great to have information on Cannabis hibiscus and what are the chances of having these species in hemp type genotypes.
Internal standard used for the study was way larger than the Cannabis genome size. Instead of Petroselinum crispum ‘Champion Moss Curled’ (2C = 4.50 pg) authors should have used multiple and similar genome size standards Primer sequences and sex determination related details are needed What was the difference between drug type and medicinal cannabis genome. Authors have also missed discussion on evolutionAuthor Response
Please see the attachment.

Reviewer 3 Report
In the current manuscript, Balant et al. gathered a large number of wild/feral, landrace, and cultivar Cannabis accessions, covering a wide distribution area in order to i) evaluate the extent of genome size and ploidy level diversity in the species, ii) investigate how this diversity distributes across accessions, geographical ranges, and sexes, and iii) test whether flow cytometry can be used as a standard tool to distinguish between male and female Cannabis individuals in both wild and cultivated accessions. Although the topic is attractive, there are some minor concerns that should be addressed.
-TITLE
The paper title is well stated, it is informative and concise.
-ABSTRACT,
Abstract needs improvement. The importance of genome size in cannabis should be explained in the introduction (it should be short but informative).
-INTRODUCTION
The introduction was well written. However, some sentences need citations.
Line 49: Please provide reference(s): (https://doi.org/10.1146/annurev-arplant-081519-040203; https://doi.org/10.1016/j.indcrop.2020.113026)
Line 99: Please provide reference(s): (https://doi.org/10.1016/j.indcrop.2020.113026)
-RESULTS and DISCUSSION
The results obtained in this study are interesting. Results presented correctly. However, the discussion of the results of “2.5. Sex Determination” is correct, but not sufficient. The discussion of this section should be improved.
Line 236: It is worth adding the study (https://doi.org/10.3390/ijms22115671) that reviewed polyploidy in Cannabis.
-MATERIAL AND METHODS
Material and research methods are presented appropriately. The experimental setup and the description in the methods section are well structured, and the statistical analysis is done alright.
-CONCLUSIONS
The conclusion is presented appropriately.
Reviewer 4 Report
The authors of the manuscript M. Balant et al. "Novel Insights into the Nature of Intraspecific Genome Size Diversity in Cannabis sativa L." did a great job in collecting samples of Cannabis sativa: a wide range of cultivars, varieties and strains growing in different areas of the world and differing in high morphological and phytochemical diversity. This was done to assess the effect of these differences on the genome size, which the authors measured by the method of flow cytometry. The main conclusion of the study is that the geographical expansion of this species and its domestication had little effect on its overall genome size. The conclusion is quite obvious, repeatedly established for a huge number of previously studied species of plants, animals, all the way up to humans. Genome size is a stable, conservative characteristic of any species with a constant number of chromosomes. That Cannabis sativa is such a species was known prior to this study. The only triploid sample in the authors' collection of 483 specimens is a rare anomaly.
Another purpose of this study was to evaluate the effect of sex on the size of the Cannabis sativa genome and the possibility of using the flow cytometry method to determine sex. From the results, the authors conclude that there are significant differences between the genome size of male and female plants. Two points remain unclear in this part of the study. Firstly, on the basis of which statistical criteria this conclusion is made? When describing the comparative analysis, the authors everywhere give delta value and percentage of differences. However, they do not indicate on the basis of which criterion the differences are statistically significant. Doubts about the statistical validity of the differences arise when considering Figure 4 (as well as Figure 1A and C earlier), in which the intervals of most measurements overlap significantly.
Secondly, it is not clear why the authors tried to use the method of flow cytometry in order to determine sex, if it was known that the level obtained by this method is very small, respectively the contribution of measurement error is high. Moreover, as it follows from the manuscript, there is a MADC2 male-associated part of the genome, which was used by the authors to separate male and female samples by the PCR method. This method is obviously more reliable and simpler than the flow cytometry.
Thus, in my opinion, of the data set obtained, only the comparison of genome sizes in female and male plants is of scientific interest, if such a comparison is described in more detail and convincingly. The scientific value of such an analysis would greatly increase if it were accompanied by a demonstration of the karyotypes of female and male plants, indicating the morphological differences in the sex chromosomes. In its present form, I cannot recommend this manuscript for publication.
